# Consciousness: A Strategy for Behavioral Decisions

**Bjørn Grinde** 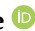

Division of Mental and Physical Health, Norwegian Institute of Public Health, N-0213 Oslo, Norway;
grinde10@hotmail.com

**Abstract:** Most multicellular animals have a nervous system that is based on the following three components: (1) sensory cells gather information and send it to processing units; (2) the processing units use the information to decide what action to take; and (3) effector neurons activate the appropriate muscles. Due to the importance of making the right decisions, evolution made profound advances in processing units. I review present knowledge regarding the evolution of neurological tools for making decisions, here referred to as strategies or algorithms. Consciousness can be understood as a particularly sophisticated strategy. It may have evolved to allow for the use of feelings as a 'common currency' to evaluate behavioral options. The advanced cognitive capacity of species such as humans further improved the usefulness of consciousness, yet in biological terms, it does not seem to be an optimal, fitness-enhancing strategy. A model for the gradual evolution of consciousness is presented. There is a somewhat arbitrary cutoff as to which animals have consciousness, but based on current information, it seems reasonable to restrict the term to amniotes.

**Keywords:** evolution; consciousness; nervous systems; feelings; reflexes; instincts; amniotes; behavioral decisions; neuronal algorithms

## 1. Introduction

No other phenomenon in nature has captured as much interest, and from such a variety of disciplines, than the question of what consciousness is about. I believe an evolutionary perspective can help inform the debate and that this perspective is best cared for by considering the role of nervous systems (NS) in facilitating survival.

NS allow an animal to behave, which generally means a coordinated use of muscles—typically to move around. The key purpose is presumably to find food, but NS also serve other purposes, such as finding mates and controlling internal organs. Plants rely on sunshine for energy and therefore do not have the same need to move.

NS have a sensory branch and an executive branch; between them lies a processing unit (Figure 1). The processing units analyze sensory input and harbor the necessary algorithms for responding to challenges by making behavioral decisions. The arguably most difficult aspect of behavior is to decide what to do or where to go; consequently, the processing units tend to be the most complex part of NS. The units have evolved from simple nerve nets to small aggregates of neurons (ganglia) and on to advanced, centralized brains. A key question for the present review is when and why evolution incorporated consciousness into the decision-making toolkit.

Vertebrates have a brain, but ganglia-like structures, such as those positioned along the spinal cord, are also involved in responses. At least two more phyla, arthropods and mollusks (annelids may also be included) evolved centralized nervous structures of sufficient complexity to warrant the term brain [1]. The importance of having an advanced processing unit is reflected in the observation that arthropods (for example, insects), mollusks (for example, octopuses), and vertebrates are by far the most successful, in terms of biomass and species variety, of the more than forty phyla present on Earth today. Large processing units offer the opportunity to utilize more sensory input, store

more information in the form of memory, and make better behavioral decisions by using more advanced algorithms.

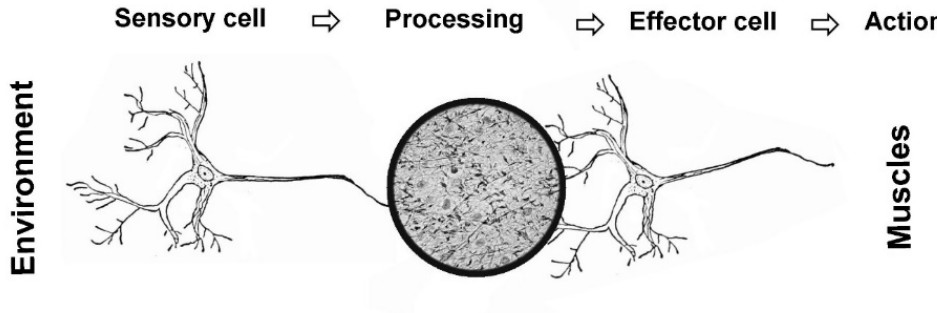

**Figure 1.** Overview of nervous systems. The majority of NS can be divided into three components: sensory cells detect environmental or internal information; this is sent to processing units (ganglia or brains) that decide on a response and execute the decision by sending signals to muscles via effector neurons.

The simplest form of behavioral response is a monosynaptic reflex, where a signal from the sensory system directly activates muscle contraction. In most situations, there tend to be many signals that are relevant, and movement depends on the coordinated contraction of several muscles; thus, evolution moved in the direction of more intricate behavioral strategies. An increase in the complexity of sensory organs and processing units evolved concomitantly with an increase in the complexity of decision-making tools. The overall purpose was survival and procreation, but this was typically cared for by equipping the organisms with more proximate targets, such as eating edible things and mating with appropriate partners [2].

I argue that to understand consciousness, it is useful to consider its role in NS. The evolutionary trajectory leading to its presence in humans is of particular interest. I start with defining key terms; then, I add some cautionary remarks, followed by a brief history of NS. In the remaining sections, I consider information pertaining to the evolution of consciousness.

## 2. Terminology

Sensory information and memory are only useful to the extent that they can form the basis for making decisions. To do so, the processing units need a strategy for computing. As an analogy to computers, the strategy can be referred to as an algorithm. The term processing unit is here used for the neurons that decide on an action and launch the response. A unit can consist of a distributed net of neurons, a ganglion, or a centralized system; it can also be absent, as in the case of monosynaptic reflexes.

An algorithm is a procedure, or set of instructions, for solving a problem; the problem for the animal is to make optimal, fitness-enhancing decisions. Consciousness and advanced cognition are the core part of the, arguably, most sophisticated behavioral algorithms. Although these options are installed in our brains, the majority of muscle activations are most likely cared for by nonconscious algorithms, for example, in the form of reflexes and peristalsis.

As to the question of who/what has consciousness, the answers in the literature range from everything in the Universe (panpsychism) to solely humans. The term is here used for a property of advanced nervous systems that allow the individual to experience life; that is, having an awareness in the form of a 'film of life.' The film may be more or less rich in content. Theoretically, it can be without content (as in a film with blank frames); but while awake, the nonconscious brain tends to supply your awareness with a stream of information.

All conscious animals presumably experience sensory input, but they are also expected to have feelings. Animals that can experience and feel are said to be sentient [3]. A feeling implies an experience with a positive or negative valence; that is, the brain delivers rewards and punishment in order to motivate behavior [4]. Rewards lead the animal toward what is good for the genes, and punishment is there to avoid what is bad for the genes.

The concept of cognition is almost as contentious as consciousness. In some traditions, it stands for any form of neurological processing. In the present text, cognition is considered a conscious process where thoughts supplement feelings in the process of finding the right action based on knowledge and intelligence. In short, cognition allows for even more fine-tuned and flexible responses. The more sophisticated forms of consciousness may also include an awareness of self. Animals that have all these tools also have a level of free will [5].

### 3. Cautionary Remarks

For a human, consciousness tends to mean a lot more than deciding on actions; thus, some people may consider the present approach an oversimplification or a reductionist view. Although I appreciate this sentiment, I believe the feature is best understood in light of the rationale that caused evolution to install early, rudimentary versions of what we have. That said, the model of consciousness I present is certainly a simplification. You need to simplify to describe natural phenomena. This is partly due to our lack of knowledge but also because nature is not constructed in a way that allows it to be accurately described by human language.

The related point is that the human vocabulary, in most cases, describes human attributes; only rarely do we coin separate terms for homologous or analogous features in animals. Thus, whether a particular term should be used when describing divergent species is a question of how different the feature can be before the term is considered obsolete.

The point is illustrated by the word 'nose.' If you ask people whether a dog has a nose, some will say 'yes;' others will say 'no, it has a snout.' In this case, there is an alternative term. In the case of concepts such as consciousness and cognition, no obvious animal-specific alternatives exist. Therefore, the question of whether other species have these attributes is a question of, for one, how different their mental capacities are compared to us; and two, how broadly we define the terms. As pointed out elsewhere, consciousness is not an either/or entity, nor simply a question of degree; each species has its own variety [6]. Obviously, it is easier to assess similarities in the case of anatomical structures compared to strategies for behavior.

We like to anthropomorphize; we see faces in clouds and human attributes in animals. As a result, we tend to believe that when we observe behavior we can relate to, the animal has experiences as we do. For example, an earthworm will twitch to escape when put on a hook, but that does not imply that the worm feels pain. One ought to be aware of this potential bias.

### 4. History of Nervous Systems

As indicated in Figure 2, metazoans probably evolved from a common, single-celled ancestor more than 650 million years ago (mya) [7]. The first cells with a vague resemblance to neurons, in the form of a capacity to signal neighboring cells, presumably appeared soon after, as we find them in all metazoan phyla [1,8,9]. Sponges (*Porifera*) were probably the first extant lineage to diverge. They are sessile filter feeders and do not need muscular movement; nutrient-containing water is passed through their pores by cellular flagella. They do not have NS, but epithelial cells can transmit signals and coordinate a response: If the incoming water is toxic, all the flagella stop beating [9]. A network of communicating epithelial cells may have been the forerunner of nerve nets, which again developed into ganglion-based NS.

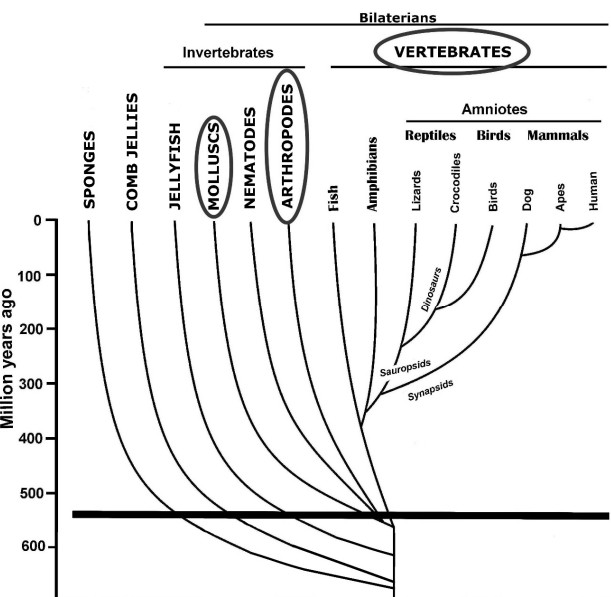

**Figure 2.** Phylogenetic tree depicting key animal lineages. Names in capital letters are phyla (trivial names are used), and the three most successful are circled. Only a few of the forty or so phyla present today are included. The thick, horizontal line indicates the 'Cambrian explosion', a time with a massive diversification of bilateral animals.

Comb jellies (*Ctenophora*) departed from the other main phyla of multicellular animals at about the same time as sponges [7]. Comb jellies have NS, but their neurons employ partly non-homologous genes and operate in a distinct fashion compared to other animals, for example, regarding the use of neurotransmitters [10]. They have a complex neuromuscular organization and a repertoire of behaviors [11]. Although basic qualities of neurons were present in our common ancestors with comb jellies, it is assumed that the resemblance of NS in comb jellies, compared to bilaterian animals, is partly due to convergent evolution [12].

The simplest form of NS do not require any processing; the sensory cells directly activate motor neurons or muscle cells, a situation seen in certain jellyfish (*Cnidaria*) [13]. They have a nerve net that integrates sensory information with muscle activation. Other jellyfish have ganglion-like structures and presumably more sophisticated strategies for orchestrating behavior [14].

Advanced NS are only present in bilaterian animals, which split off from radially organized animals, such as jellyfish, soon after the split with comb jellies. Only a few bilaterian phyla moved on to brain-like structures, and it is likely that this, too happened through convergent evolution [15]. Centralized NS offer the obvious advantage of more advanced processing and coordinated response for the entire body, which explains why this development could take place independently in at least three lines of descent. A brain is presumably a prerequisite for consciousness, but advanced neurological processing can take place in the absence of consciousness.

All NS (apart from those of comb jellies) are built on neurons with similar characteristics as to the core genes involved, axonal transfer of signals, and transfer of signals between neurons [1,16,17]. In other words, key neurotransmitters, such as serotonin, dopamine, and opiates, are likely present in all bilaterians and tend to serve partly homologous functions [18]. Although the more diverse (and chemically elaborate) peptide neurotransmitters are also present in most NS, their variety and use are drastically expanded in complex systems such as those of mammals [19]. The mammalian NS are also characterized by having the largest number of neurons (and their support cells), more connections (synapses) for each neuron, as well as a greater variety not only of neurotransmitters but also of the proteins the neurotransmitters interact with.

## 5. Nonconscious Algorithms

As pointed out above, even sponges can create a coordinated response to environmental stimuli. So can plants: in the shrub *Mimosa pudica,* all the subparts of a leaf fold if touched at one spot. Although these examples can be construed as a form of behavior, I reserve the term for muscular responses orchestrated by NS. Behavior does include responses that are invisible to someone observing the animal, such as the contraction of muscles surrounding blood vessels or the gut. This way of defining behavior may differ from typical daily use but is preferable when analyzing the algorithms of NS.

Monosynaptic reflexes are the simplest form of behavior. Even advanced NS, such as the human version, use this type of reflex, as exemplified by the knee-jerk (patellar reflex). The knee-jerk does not require any processing; neurons sending a proprioceptive signal from the quadriceps muscle directly activate motor neurons based in the spinal cord that cause the same muscle to contract. The knee-jerk is part of a nonconscious algorithm designed to help us retain body posture.

Reflexes are not necessarily monosynaptic, but they generate a response with minimal processing. Their speed and simplicity presumably explain why this algorithm is widespread in animals. A fixed action pattern is similarly stereotyped but is typically used in more complex situations. That is, the response takes more information into account, for example, in the form of diverse sensory input, and requires the coordinated action of several muscles.

Reflexes and fixed action patterns deliver stereotyped responses that are not designed to be extensively modulated. The next evolutionary advancement was to allow for modulation based on previous experience, what is broadly known as learning. As it is difficult to preprogram responses to the enormous variety of situations an animal is likely to encounter, decisions based on previous experience offer obvious advantages. Modulating the architecture of the NS in this way does not require a complex system; even the nematode *Caenorhabditis elegans*, with its 302 neurons, is capable of learning [20]. However, the relative importance of learned, rather than fixed, responses is likely to increase with the complexity of NS.

The term reflex is sometimes used for reactions that are subject to learning. Humans can respond to danger with a reflex, such as a startle response, when spotting anything resembling a snake. A person who is familiar with and appreciates snakes is less likely to experience any startle. The point illustrates that with a brain as advanced as ours, even basic algorithms can be subject to modulation and interference by mental activity.

Another useful algorithm is of selective attention. In the simplest NS, there is presumably no filtering; all sensory signals have the same chance of eliciting a response. It is useful for an organism to focus attention on signals that are likely to be of particular importance. Even nematodes have this capacity. They can, for example, selectively enhance signals, such as the smell of a bacteria they feed upon, and thus direct movement more precisely in the appropriate direction [8].

Instincts and instinctive tendencies are more complex patterns of behavior. They imply that the organism has an inherent inclination to a particular form of response, but the response can vary considerably between individuals because it is modulated by learning and subject to genetic variation. Instincts can operate in the absence of consciousness, but in humans, the accompanying behavior is typically brought to conscious attention and thus more or less available for interference.

In associative or conditional learning, the individual form an association between one event and another. One event is typically non-consequential, while the other has a positive or negative fitness value. The association means that the individual will prepare for the second event when encountering the former. This form of learning has been demonstrated in several invertebrates, for example, sea slugs [21].

In mammals, conditioning is associated with feelings. The animal repeats an action because it obtained a reward on a previous occasion. Although conditioning is unlikely to

require consciousness, in cases where it is based on rewards or punishment, the process is expected to be conscious.

Terms such as imitation, insight learning, intelligence, and cognition tend to be associated with conscious processes, but the terms are also used to describe behavior in species that may lack consciousness. They offer relevant descriptions for advanced behavior in animals such as bees and octopuses, but if the present position that these animals are nonconscious is correct, the responsible algorithms are necessarily different from the corresponding algorithms in humans.

There is considerable knowledge as to how the neurons of *C. elegans* enable these worms to learn and make decisions [22]. Moreover, there is no need to implicate consciousness to explain their behavior. In the more advanced NS, the algorithms associated with reflexes, fixed action patterns, attention, learning, and instincts can lead to very complex behavior, but again consciousness is not necessarily required.

Evolution has moved in the direction of highly intricate responses that involve several consecutive procedures. The algorithms responsible need to coordinate the activity of a large set of muscles over a prolonged period. One example is the dance of honeybees used to communicate the direction and distance of a food source. The behavior depends on learning, as it needs to be adjusted to the specific location communicated each time; yet, it is highly stereotyped, as witnessed by the observation that different species of honeybees have distinct dialects [23]. In my mind, this form of behavior most likely does not require sentience; it can be accounted for by an elaborate, nonconscious algorithm.

Even brains with the capacity to make conscious decisions tend to retain the various forms of nonconscious algorithms. The observation suggests that evolution is constrained by a trade-off: elementary algorithms are faster and require less energy (both in the establishment of the NS and their operation), while complex algorithms offer increased flexibility and versatility. The choice depends on the behavioral requirements of the species and on the task at hand. Consciousness is the most versatile but also the most expensive algorithm. The trade-off can explain why, even in humans, it is only employed for select purposes. That is, the brain sends information to your awareness on a 'need to know' basis.

Most of the processing required to keep the body going is nonconscious; conscious decisions are only called for in situations where they offer distinct advantages. The limited utility of consciousness is an important observation when considering which species are likely to have this capacity. One would expect it to evolve only in species with a particular need for behavioral flexibility.

Humans are made aware of a select subset of the responses that are based on nonconscious algorithms. We are typically reminded when voluntary (striated) muscles are involved but not the activation of non-voluntary (smooth) muscles. For example, we are aware of the knee-jerk, but we are unaware of activity in the iris, a smooth muscle surrounding the pupils. Its contraction upon increasing light is based on a reflex type of action.

## 6. The Disadvantages of Consciousness

We tend to think of consciousness as the ultimate gift to life on Earth. True, without it, we would not have science, and we could not experience or feel anything. The higher form of consciousness and cognition offers extreme versatility and adaptability; yet, in terms of biological fitness (survival and procreation), it is, perhaps somewhat surprisingly, not such a splendid attribute.

Arthropods (*Arthropoda*) are the most successful phylum of animals both in terms of species variety and biomass. The most successful species in terms of biomass belongs to this phylum; a tiny shrimp, the Antarctic krill (*Euphausia superba*), is assumed to represent twice the biomass of humans [24]. Although some claim that arthropods have consciousness [25], their brains, and correspondingly their putative conscious resources, are minute compared to mammals.

The lineage with presumptively the most sophisticated form of consciousness, hominids, had limited biological success based on the above criteria until recently. Going back some 100,000 years, there were probably five other species of Homo (*H. erectus*, *H. heidelbergensis*, *H. naledi*, *H. floresiensis*, and *H. luzonensis*) and three subspecies of *Homo sapiens* (Neanderthals, Denisovans, and 'modern' humans). All but one of these lineages are extinct. Even our line of descent went through an evolutionary bottleneck at one point [26]. Moreover, the apes, our closest relatives, are not doing very well; some of them, including the mountain gorilla and the bonobo, are threatened by extinction. In short, even a highly intelligent and advanced brain does not guarantee success.

The explanation may be related to the trade-off discussed in the previous section. Consciousness is a costly and slow strategy for making decisions. The human brain demands some 20% of our energy, although it only stands for 2% of body mass. The sluggishness is illustrated by the fact that a reflex can be executed in less than 20 ms, while simply the perception of a sensory stimulus takes some 300 ms [27]. Conscious decisions typically take a lot longer, as perception (or retrieval of memory) is only the first step. The time spent presumably correlates with the energy required. Brain resources are precious and limited, as witnessed by the observation that less important functions tend to degenerate to allow for more important functions. For example, olfaction has been down-prioritized in the human lineage, presumptively to enhance other functions such as vision and cognition [28]. The point reflects the general principle that the mass of neural tissue required for a particular function reflects the amount of information processing involved in performing the function [29].

Consciousness only focuses on one task at a time. If additional tasks require attention, the brain shifts back and forth between them [30]. Nonconscious processing can simultaneously care for multiple assignments, such as regulating heartbeat and pupil size.

The conscious brain lacks the power to execute its decisions, such as orchestrating the muscles of legs when walking [31] or talking [32]; the latter requires fine-tuned coordination of more than two hundred muscle movements every second. We believe we master both walking and talking, but all we do is decide where we want to go and what we want to say; the extremely complicated use of muscles is cared for by nonconscious algorithms. The point is reflected in the observation that we can both walk and talk while asleep. Most people find it difficult to get around without seeing, but that may not be due to a need for conscious control but rather the need to keep eyes open. That is, visual input must reach the brain, but it does not need to be consciously processed, which presumably is the case when sleepwalking.

In short, consciousness is for making decisions; executing them requires coupling with other brain resources, while in nonconscious algorithms, this is presumably one streamlined process. Finally, conscious decisions are vulnerable to (biologically speaking) dangerous whims. At least this is the case in humans: we may decide not to have children or even to kill ourselves.

Based on the above discussion, it seems fair to state that consciousness is not the ultimate fitness-enhancing innovation.

## 7. Ostensible Consciousness

When we see an animal react to something we conceive as painful, we assume that the animal feels pain. The earthworm twitching on a hook is an example. All animals react to noxious stimuli simply because avoiding injury is a core function of NS, but there is no need for the reaction to be conscious. If you inadvertently put your hand on a hot stove, jerking it back is a reflex. Awareness and pain come later. The algorithm controlling the reaction is nonconscious simply because conscious decisions are too slow for this type of situation. There is no need to postulate conscious feelings in the earthworm when it performs a related action. It is also possible to be aware of a harmful situation without sensing anything, as in people with an innate inability to experience pain.

The startle response upon seeing something resembling a snake is a similar case. The sensation of fear appears after the startle. The importance of avoiding danger has led evolution to settle for a nonconscious algorithm controlling the initial reaction; the subsequent fear is useful as a learning experience. We use cognition to evaluate the situation after it happens, but the key part of the response is nonconscious.

All animals are expected to have algorithms that help them avoid danger. If the response can be cared for by nonconscious processing in humans, the same is likely the case in other species. The observation that fear-related behavior in crayfish can be prevented by a (human) anxiolytic drug (a benzodiazepine) was considered an indication of feelings, and thus consciousness, in these arthropods [33]. However, benzodiazepines boost the effect of the inhibitory neurotransmitter gamma-aminobutyric acid (GABA), and this neurotransmitter dates to the early NS and tend to have conserved functions. It calms down neurological activity. In other words, this sort of observation is expected if an invertebrate nonconscious response employs the same neurotransmitter as the related, feeling-associated response in humans.

Nonconscious processing is most likely more prevalent than most people realize. We do, for example, base decisions on sensory input that we are not aware of, as in the case of subliminal perception [27]. If an image is flashed for 50 ms, the person seeing the image will not be aware of the content but can still respond as if he/she did see it. Some sensory-based control systems do not involve consciousness at all, for example, the regulation of temperature. There is an indistinct line separating the conscious from the nonconscious.

Nervous systems are not computers, but if a strategy for making decisions can be programmed in a computer, I would expect that it is possible for evolution to design kindred, nonconscious algorithms in a brain. Moreover, based on what I have outlined so far, nonconscious algorithms have distinct advantages compared to conscious alternatives. To understand consciousness and have an educated opinion as to which organisms have this property, it is helpful to consider what cannot be achieved in its absence.

### 8. Signs of True Consciousness

For evolution to install consciousness, there needs to be a distinct advantage. The main function of NS is to orchestrate behavior; the obvious benefit is to offer increased versatility and flexibility in behavioral decisions. As pointed out above, complex responses can be dealt with by nonconscious processing; consciousness adds a level of adaptability that may be difficult to program even with the tools mentioned so far. It allows the beholder to find a solution to novel situation without previous related experience. In other words, consciousness offers a particular advantage if relevant factors are difficult to anticipate (and thus program), for example, because they vary considerably in strength and character. This is the case for animals in a complex and dynamic environment.

Consequently, when assessing for the presence of consciousness in a species, one should look for signs of versatility, such as behavior that differs both between individuals and when the same individual faces related challenges. Unfortunately, it is difficult to implement this criterion, as we do not know the actual power of nonconscious processing.

There seems to be some consensus in lining up candidates that might possess consciousness (in decreasing order): mammals, birds, reptiles, amphibians, fish, octopuses, and certain arthropods such as bees [6,34]. Below, I present information that can be used to postulate which of these animals are likely to possess anything resembling our consciousness.

### 9. The Role of Feelings

One line of investigation is to try to trace the evolutionary trajectory leading to consciousness. The approach is like tracing the process that led to eyes in vertebrates. Both eyes and consciousness seem to be an either/or quality, yet the features necessarily evolved stepwise from nothing to full-blown versions. In the case of eyes, the starting point was photosensitive cells in the skin that allowed the organism to orientate according to light

intensity [35]. The capacity to respond to light gradually improved until some organisms were able to form internal (nonconscious) images resembling their surroundings.

Based on the above, it seems prudent to ask why a rudimentary capacity to experience something was an advantage. Both sensory information and highly complex behavior can presumably be cared for by nonconscious algorithms, and as pointed out in previous sections, consciousness has disadvantages, so the answer is far from obvious. The observation that the capacity to experience life seems to be coupled with the presence of feelings may offer a clue [36,37].

Feelings reflect an algorithm where behavioral decisions are based on using positive or negative valence as a 'common currency' for weighing options [38]. In other words, the individual tries to maximize the output of positive feelings. If, for example, the animal wishes to drink from a source of water, but a predator lurks on the shore, it is a question of weighing the pleasant anticipation of quenching one's thirst against the unpleasant fear of being attacked. Several factors are relevant for evaluating alternatives, such as the state of dehydration and the availability of weapons for defense. These factors modulate the strength of perceived positive and negative valence. Feelings offer a versatile algorithm, as they can be used for a large range of situations. They are also adaptable in that the strength of the feelings is modulated by several factors—including previous experience.

The weighing does not need to be conscious, but the animal needs the capacity to feel pleasure and pain. This could be the incipient step in the direction of consciousness. Feelings may have evolved from nonconscious motivators set up to impact behavioral decisions. Feelings require the capacity to experience; once an animal had this capacity, it may have been easy to add other forms of experiences, such as sensory input. It seems pertinent that the organism should experience the stimuli that give rise to the feelings. These two types of mental content (reward/punishment and sensations) are combined in full-blown sentient animals and constitute the basic form of consciousness. If this idea is correct, feelings initiated the evolutionary process leading to conscious beings. Their impact on behavioral decisions offered the evolutionary advantage—consciousness can be construed as a by-product.

One argument in favor of the above model is that feelings may represent the first algorithm that requires a capacity to experience. Human cognitive deliberations depend on consciousness, but they are likely a more recent enhancement of our conscious capacity.

For a human, the most manifested experiences associated with consciousness tend to be those provided by the senses, particularly vision and hearing. However, it is difficult to envisage this sort of experience as the root, or evolutionary rationale, for consciousness. Even the most elementary NS respond to information coming from sensory cells, and since then, sensory organs have evolved gradually. One may hypothesize that sensory information at one point became so complex that consciousness would take over from nonconscious algorithms in processing and responding to the information, but for me, there is no obvious reason why that should be the case. Handling complex information is not the strength of conscious processing, which is why the highly demanding task of orchestrating muscles for the sake of walking and talking is, to a large extent, left to the nonconscious. A computer, or the brain of a bee, can be programmed to analyze and respond to complex optical images. In fact, even humans base their behavior on the nonconscious processing of complex visual or auditory stimuli when the stimuli are subliminal, or the person is sleepwalking. Typically, conscious and nonconscious processes work together to execute elaborate tasks.

Not only can we act on nonconscious sensory information, but we can also respond to emotional stimuli without being aware of the emotional impact [39]. Both situations may reflect relics of processes in pre-conscious animals. The observation that we can respond to sensory and emotional information in the absence of conscious recognition is in line with the idea that the capacity to experience this information evolved gradually from nonconscious algorithms.

## 10. Neurological Considerations

Consciousness may not require a particularly large brain. Assuming that all mammals have this capacity, the 60 mg brain of the dwarf shrew [40] is sufficient. If these animals were to lack even a minimal form of consciousness because the feature requires more neuronal tissue, there would need to be a cutoff. As the size of adult mammalian brains forms a continuous scale from 60 mg to 8 kg (the latter belonging to the sperm whale [41]), a cutoff seems unlikely. A more parsimonious scenario is that consciousness itself does not require extensive neuronal resources, but that size limits the amount of information the brain can handle and the capacity to analyze the information.

In comparison, *Octopus vulgaris*, a medium-sized octopus commonly used in research, has a brain of 2 g [42], which is well above that of the dwarf shrew, but, according to the present theory, octopuses are not conscious. As expanded on below, it seems likely that neuronal interconnectedness is a more relevant requirement than size [43–45].

Consciousness seems to depend on having many neurons linked together in a complex network. Mammalian awareness apparently relies on the capacity for continuous back-and-forth signaling in the interconnected circuits of the thalamocortical complex [34,46]. This complex is referred to as a global neuronal workspace; the relevant neuronal circuits are capable of global activation and itinerant activity [27]. There is incessant basal firing in the absence of awareness; the idea is that a sentient experience implies distinct perturbations in the basal signaling. This change of activity can be observed in electroencephalography (EEG) patterns in mammals and is considered to signify consciousness [47,48].

Transcranial magnetic stimulation combined with EEG can be used to evaluate whether individuals in a coma or related states have retained a capacity for consciousness [49]. The stimulation implies a 'knock' on the head. If the knock induces long-range, complex signaling, it is taken as a sign that the brain has the required neuronal excitability.

The requirement for complexity in signaling is also covered by the integrated information theory [50]. According to this theory, the level of consciousness depends on the complexity of integrated information a system, such as a brain, can handle.

The cerebellum contains approximately 80% of the brain neurons but is presumably not directly involved in consciousness. The organ is important for the coordination of muscle movements, the number of neurons involved suggests that this task is more demanding than making the conscious decision to move. The neurological activity of the cerebellum seems to be based more on feedforward signaling [51], rather than the extended back-and-forth signaling associated with conscious processes in the thalamocortical complex. The observation supports the idea that it is not the number of neurons but the way their activity is connected that distinguishes consciousness.

Apparently, the most highly interconnected neurons are in the mammalian cortex and the associated forebrain structures, such as the thalamus. The long-distance interconnectedness of neurons seems less developed in other vertebrates and even less so in invertebrates. The point is presumably reflected in the observation that only mammals have large domains of white matter [52]—areas of the brain filled primarily with myelinated nerve fibers. The cortical white matter contains the axonal projections of neurons in the outer grey matter. The gathering of long-distance nerve fibers in separate regions may be part of the evolutionary change required for mammalian consciousness, as it allowed for an increase in long-distance interconnectedness.

Surprisingly, consciousness appears to be possible even in the absence of a cortex. The evidence is based on mammals where the cortex is removed [46,53], and on children with hydranencephaly [54]. These children are born without cortex or with minimal remnants thereof, yet they not only can survive but show signs of emotions and thoughts. It is conceivable that the remaining structures of the forebrain are sufficient to generate a basic form of consciousness.

## 11. Invertebrates—A Question of Convergent Evolution

Octopuses are considered the best candidates for consciousness among invertebrates [34]. These animals can, for example, learn to navigate complex mazes and utilize tools [55]. Moreover, they can apparently use future expectations to modify behavior: if shrimps are likely to become available later, they will eat less crab now [56].

While these forms of behavior suggest considerable versatility and flexibility, it is not obvious that they are outside the scope of nonconscious algorithms. Yet, the main reason why I do not expect that octopuses, or other invertebrates, to have consciousness is that it would require convergent evolution. As discussed below, the convergent evolution of this trait seems unlikely. If they had a related trait, I would expect it to be too different from human consciousness to warrant the use of the term. It should be kept in mind that the common ancestors of arthropods, mollusks, and vertebrates had simple and presumably nonconscious NS (Figure 2).

Convergent evolution is a well-known phenomenon. One example is the wings of species that have adapted to flight. However, the superficial resemblance of wings in birds and insects can be explained by the physical requirement for flight. Their wings are as similar as they need to be for the purpose. Wings of birds and bats are more alike, but that is because they both evolved from limbs present in their shared ancestors.

The convergent evolution of NS (in comb jellies and bilaterian) and of brains (in phyla such as arthropods, mollusks, and vertebrates) is somewhat like wings. Behavior (beyond simple reflexes) depends upon a processing unit, and advanced behavior requires reasonably large, complex, and centralized NS. Moreover, the starting point was shared in the form of assemblies of neurons forming either nerve nets or nerve cords.

Advanced decision-making algorithms do not have any obvious need to converge; presumably, there are many strategies that allow for complex responses, as there are many ways to program similar tasks in a computer. The shared ancestors of the relevant phyla most likely did not have any properties that would point in the direction of feelings (or consciousness) as a strategy. The above discussion on the advantages and disadvantages of consciousness does not suggest any compelling reason to move in this direction.

While we cannot rule out some form of awareness in invertebrates, this feature is probably not required for the behavior observed. I have pointed out examples of perhaps equally advanced, nonconscious processing and decisions in humans; thus, invertebrates should be able to cope without them.

Substantiated by the discussion so far, I hypothesize that consciousness is only present in vertebrates. The next question is whether it is present in all or only a subset of these animals.

## 12. Vertebrates

While we can only be sure of our own consciousness, its presence in other mammals seems likely. Their brains have reasonably similar anatomy, the neurological and behavioral correlates of consciousness are present in the species examined, and there are definite signs that they use feelings as a tool for evaluating options [37,50,57]. Signs of self-awareness are found in certain species, such as apes and cetaceans [58]. While one can be conscious without self-awareness, the latter is probably a suitable indicator of consciousness as it is somewhat easier to demonstrate than the presence of feelings.

Birds, and to a lesser extent reptiles, have brain structures and activity typically associated with consciousness, while these features are less developed in amphibians and fish [34,59,60]. The avian pallium (which corresponds to the cortex) and thalamus support itinerant activity similar to that found in the corticothalamic complex of mammals. Neuronal responses to visual stimuli in corvid birds are comparable to those associated with visual awareness in humans and have therefore been considered an empirical marker for a conscious experience [61]. In other words, birds seem to have neuronal hardware comparable to that assumed to generate consciousness in mammals [62].

The transitions from reptiles to birds and from early amniotes to mammals entailed a considerable increase in brain size. The increase was mainly in the forebrain and is

dominated by the pallium of birds and the cortex of mammals [60]. Various studies suggest that species of the corvid and parrot orders have behavioral abilities on par with those of primates; for example, as to the use of tools, future planning, causal reasoning, and imagination [63–65]. There are also signs of self-awareness based on the mirror test in birds [66,67].

Given that birds and mammals have consciousness, the next question is whether the capacity is present in reptiles. The alternative is that it is involved independently in the former lineages. An assessment of neurobiology reveals several signs of convergent evolution in birds and mammals [68,69]. Both lineages apparently evolved brains better equipped to cater to consciousness compared with what we find in reptiles. Extant lineages of reptiles are not as advanced in terms of behavior as birds, yet there are signs of consciousness. Some reptiles are capable of complex forms of learning, such as social and associative learning, as well as problem solving [70–72]. Although the signs are less distinct compared to birds, it seems likely that evolution started with a brain possessing a rudimentary form of consciousness rather than inventing it independently in birds and mammals.

It is less obvious that fish and amphibians are conscious. Two important lines of evidence backing this statement are based on the idea that consciousness started with feelings: One, dopamine serves a key role in both the reward part of feelings [73,74] and in consciousness [75]. It has been suggested that the substantial increase in telencephalic dopamine receptors in amniotes, as compared with amphibians, supports the notion that both consciousness and feelings evolved after these lineages split [76]. Two, while reptiles show physiological signs of feelings, such as tachycardia (rapid heartbeat) and increased temperature upon handling, these signs are not observed in fish and amphibians [76].

There is additional evidence backing the idea that fish do not have feelings [77]. Moreover, although amphibians and fish have highly complex behavior, it seems less versatile and more within the range of what can be programmed.

Diurnal cycling can be observed in most organisms, but REM sleep is only present in certain animals. REM sleep is associated with dreaming. Based on EEG patterns, it is a state that resembles awake awareness. (Some claim that REM sleep is a form of consciousness. This is primarily a question of definition. My stance is that REM sleep should not be framed as a conscious experience as the individual is (normally) not aware of what is happening and is not in control.) The distinct changes in EEG, between slow oscillations associated with deep sleep and the conscious-like oscillations of REM sleep, are consequently considered to indicate a capacity for consciousness [34]. Previously, REM sleep was presumed to be restricted to birds and mammals [78], but a similar state has recently been observed in reptiles [79,80].

To summarize, various lines of evidence suggest that some form of consciousness is present in amniotes (reptiles, birds, and mammals) but not in fish and amphibians. The amniote lineage diverged from amphibians some 310 million years ago [81], which means that the feature started to evolve after that date.

## 13. Why Amniotes?

The disadvantages of consciousness suggest that it only evolved for situations where nonconscious algorithms are less suitable, which presumably implies environments that are rapidly changing, unfamiliar, and complex. Feelings and consciousness reflect a strategy for obtaining the required flexibility of behavioral response. The following points support the idea that the process started in the amniotes:

1.  The amniotes were the first vertebrates to colonize land fully, and this adaptation was accompanied by significant neural expansion [8,60]. Large brains implied a suitable foundation for more advanced algorithms.
2.  They evolved lungs to breathe air, which allowed for an increase in available oxygen (and thus energy) compared to gills. This is a likely requirement for expanding the brain.

3. Amniotes occupied a novel habitat; that is, land rather than the ocean. This habitat was much more variable regarding important features such as temperature, water availability, and seasonal changes.
4. Land as a habitat offered a range of novel niches, a situation that typically corresponds to rapid evolutionary change.
5. Arthropods colonized land before vertebrates, but the early amniotes were larger, lived longer, and typically had smaller litters. This combination means they could not adapt rapidly by genetic evolution. Instead, each individual needed to adjust to environmental challenges within their lifetime; in short, they required flexible behavioral algorithms.

## 14. The Ontogeny of Consciousness

Padilla and Lagercrantz [82] suggest that the emergence of consciousness in fetuses and infants follows the same progression as in phylogeny. If so, and based on the present model, the capacity to respond to sensory input should come first; and the capacity to feel should precede actual (sensory) awareness. Response to input from sense organs starts to develop at about 10 weeks of gestation [82]. The earliest response is based on touching the lips; other senses follow, so at birth, all key sense organs are functioning. As discussed above, responding to sensory information does not require the conscious perception of the stimuli. In fact, the fetus responds to stimuli but is not aroused, which suggests it is in a nonconscious state [83].

The brain displays EEG activity akin to that of awake adults long before birth. The thalamocortical connections assumed to be a key feature in relation to conscious processing first appear at around 24 weeks of gestation [82], but they are not fully developed until after birth. However, similar EEG activity is observed during REM sleep, and fully developed thalamocortical connections are present in adults, whether they are conscious or in a coma. Thus, the presence of these features does not prove consciousness.

Birth seems to represent a form of awakening. The newborn soon starts to display what appears to be emotional responses, yet the newborn may not be truly conscious. The smile, which is typically considered a sign of emotions (and thus consciousness), appears spontaneously as a reflex [84]. The various features of adult consciousness may develop gradually after birth, and if feelings precede true awareness, the baby may start to experience some form of joy when smiling soon after birth. Similarly, crying may be nonconscious but still reflect a form of pain.

In adults, verbal report is typically used to probe the level of consciousness, but this is not possible with babies, which means we do not know when sufficient awareness is in place to refer to the child as conscious. I believe the more telling data come from experiments with event-related potentials. This signature of brain activity appears when a visual image is presented for long enough to reach awareness (rather than being subliminally processed) [27]. Experiments with babies have shown that the event-related potential develops from a weak and delayed response in babies of 5 months to a response more like adults at 15 months [85]. None of the behaviors observed in the first year of life require consciousness but based on these experiments; my guess is that the capacity starts to develop at birth and that a reasonable level of awareness is reached by the age of one. Others consider the responses observed in fetuses as signs of consciousness [86].

## 15. Conclusions and Prospects

I believe the evidence favors a model where the evolution of consciousness started some 300 million years ago, and the feature is present in reptiles, birds, and mammals. The same conclusion was made based on a rather different theory as to what consciousness is about, the attention schema theory [87,88].

Any complex trait needs to evolve gradually. The present model suggests the following scenario: behavioral motivators turned into feelings, feelings gave the capacity to experience, more features (including sensory information) were added to the repertoire

of what an organism can experience, and slowly the capacity turned into higher forms of consciousness including cognition (in the present sense), intelligence, and self-awareness.

While I am comfortable using the term consciousness for all mammals, stretching the concept to birds and reptiles is less obvious. I do believe these animals have features resembling consciousness, but their 'film of life' most likely differs considerably from ours. The question of who is conscious depends on where we set the cutoff for the use of this concept. It is conceivable that a rudimentary form was present even in amphibians and possibly fish. Our knowledge is too limited to be sure of a valid cutoff; however, placing it at the amniote stage seems reasonable.

In the case of invertebrates, I consider it less likely that the concept of consciousness is appropriate. Octopuses have highly advanced behavior, but it is probably within what can be programmed. It is possible that they evolved some 'conscious-related' algorithm, but if so, I expect their algorithm to be very different from what we have. Too different to deserve the term consciousness; for example, it may be based on something entirely different from feelings. Evolution does not work toward optimality, only toward survival. Thus, consciousness is not necessarily the best strategy for behavioral control. Perhaps evolution just 'happened to stumble upon feelings,' and this feature proved sufficiently beneficial to allow the owners to prosper. The octopus lineage may have evolved an even better algorithm.

For humans, consciousness is a lot more than making decisions based on feelings, but our intellectual capacity came later and was most likely not the driving force behind the initiation of consciousness. We are today extremely successful mammals, but our success started long after we obtained the genetic constitution of our species. Consequently, the success is probably due to cultural evolution rather than our biological inheritance.

I believe the model of consciousness I present has merit, but it is in dire need of more evidence. One possible line of investigation is to enhance our understanding of the neurological correlates of consciousness. One intriguing observation is the difference in neurological signature when visual information is experienced as either subliminal or conscious [27,89]. Typical experiments use pictures flashed for varying duration, but it is possible to employ other sense organs. It would be interesting to compare responses in fish, amphibians, reptiles, birds, and mammals. We cannot ask these animals what they experience, but it should be possible to tune the conditions (for instance, the duration of a sensory presentation) so that one can differentiate between a limited (subliminal) activation in the brain or a global activation suggestive of consciousness. Based on the present model, that difference should be visible in amniotes. The approach has been shown to be feasible when studying human babies [85].

Another interesting option is to look at which animals will self-administer drugs that stimulate the reward function of the brain. Mammals will typically seek drugs affecting the reward system (such as heroin, cocaine, and alcohol) but not hallucinogenic compounds [90]. The latter only offers indirect rewards for a particularly curious and explorative species. To my knowledge, the self-administration of drugs has not been studied in invertebrates or non-mammalian vertebrates. Data would be unreliable, particularly for invertebrates, as they could have reward systems that are not affected by drugs known to humans; but as mentioned above, anxiolytic drugs seem to have an effect on crayfish 'anxiety' by acting on GABA circuits [33]. The typical drugs of abuse act on dopamine or opioid circuits, and as in the case of GABA, these neurotransmitters are present in most animals.

Consciousness is not necessarily the most amazing evolutionary achievement. In fact, our nonconscious brain can care for many of the tasks required to respond to environmental challenges, as witnessed by subliminal perception and sleepwalking, which suggests that similar feats performed by other animals do not need to imply the presence of consciousness.

Conscious experiences are only the ripples on the ocean of human brain activity. Yet, without the capacity to experience life, we would have nothing.

**Funding:** This research received no external funding.

**Institutional Review Board Statement:** Not applicable.

**Informed Consent Statement:** Not applicable.

**Data Availability Statement:** Not applicable.

**Conflicts of Interest:** The author declares no conflict of interest.

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
