# Peer review of "Consciousness: A Strategy for Behavioral Decisions"

_encyclopedia, doi:10.3390/encyclopedia3010005_

Round 1
Reviewer 1 Report
The authors should address the ontogeny of consciousness in humans beginning during fetal life. Either they should address changes in the level and content of consciousness with prenatal and postnatal maturation or state up front for the reader that this is an evolutionary discussion without, including the unique context of human consciousness. However, this omission is quite important to correct before excepting this manuscript.
Interesting work by the Swedish neonatologist Hugo Lagercrantz in Early Human Development 2009 would be a useful starting point to consider adding this discussion. The same author reviewed this topic more broadly in Pediatric Research 2009.
Author Response
The question of when in human fetal/early life consciousness starts to appear is interesting and highly relevant. I am aware of the work of Lagercrantz. I felt the topic was a bit beyond the scope of the article, but I am happy to include a section on The Ontogeny of Consciousness in the revised version.
Reviewer 2 Report
The present review is collaging various argumentations regarding consciousness in animals, starting from terminology, some history of the nervous system, the existence of reflexes and non-concsious processes, the cost of consciousness, signs of true and non true consciousness, the potential role of feelings as a basic coding language, neurological considerations, the question on where between vertebrates and invertebrates the concept stops, and the conclusion.
General comments
The article in itself is well written, by an author whose previously published work is known. Most of the argumentation is also not wrong, but the true point eludes me.
Firstly, I would like to how this work is different from for example
Grinde, B. (2018). Did consciousness first evolve in the amniotes? Psychology of Consciousness: Theory, Research, and Practice, 5(3), 239–257. https://doi.org/10.1037/cns0000146.
Or the 2016 book from the author as well (The evolution of consciousness, Springer).
Secondly, I do not see the harm in most of the arguments developed, but rather how they constitute a new model
Thirdly, throuhgout the manuscript, consciousness is purported as a binary feature. There is or there is not consciousness; which is intrinsicly incoherent with the author's own argument that the evolution of most traits is necessarily progressive. In the very same way, animals with the highest consciousness probably retained lower forms of consciousness as well. The author sees, as many, the linearity of evolution and portraits consciousness as a feature, acquired, and then discusses its costs and benefits. I think this is an all too clean way to represent this messy process. Consciousness to me, like other traits, is an emerging feature, given sufficient network and signalling capacity. I would like this discussed furter.
Specific comments
Abstract
L15: why does the author think that in biological terms it does not seem to be an optimal, fitness-enhancing strategy? How much more can a bald light primate enhance its chances of survival?
L71: Algorithms are not originating from or limited to computer science.
L79: I'd say that visceral movements such as respiration or peristaltism could be aptly used as examples here.
L81: what sort of literature? Philosophical? Cognition? Psychology? Neuroscience?
L105: I would prefer wording to be along the lines of 'the human language can only describe nature to a point' rather than nature was designed. I understand the point is the same though.
L177: what is the rationale behind this reserve? The reaction of a plant to changes or challenges in its environment are not coordinated? What about acacia bushes communicating that a giraffe is eating them to its conspecifics, which then produce a bitter molecule that the giraffe does not like? Terms do matter, as the author points out, therefore behavior needs boundaries, and the topic is the nervous system, but there is a little bit more to extract from this plant example, and a little bit more justification as to why the term behavior is restricted to NS-beings.
L255-259: the point here is that consciousness is called only where necessary, where there is an advantage. Let's consider that a human is hungry. This hunger is likely very similar to that in other animals, and a shared trait, the need for food. The human will be conscious of that, a given mammal such as the dog likely as well, but if we go all the way to non-conscious mammals, there is a point at which consciousness is necessarily not involved. Is that because mammals require complex behaviors to feed? Because humans need to hunt? I do not think so. A lot of feeding behaviors are very repetitive and similar in a lot of species, yet we are aware and conscious of hunger. I would like to see this part slightly developed.
L277-284: this is not a very good argument. First, homo erectus arguably spread to the rest of the world from a subregion of Africa, then yes, other hominid species eventually wiped it out, but hominids from that point of view were extremely successful compared to a lot of other species, including arthopods, which may be very numerous but rarely are present all over the globe. Second, biomass is one thing, but the threat of extinction in a world we have more than exponentially altered is poised to pose a threat to our closest lineages along with all the species living in forests and jungles. The European wolf is not doing very well either, while the European rat is thriving, which does not prove anything specifically. In this interviene factors such as gestation period and litter numbers for example, or diet flexibility which is not correlated to higher thinking.
L320-326: The author intentionally mixes pain and consciousness. Consider the genetic disease that is a mutation in the SCN9A gene, which causes humans to not feel pain. They are still conscious (obviously), yet do not feel pain. Pain, nociceptors and their pathway are a part of a wide range of animals NS. Therefore, behaviors we associate with pain may not imply consciousness, but may very well and often do, imply pain.
L343: there are grades of consciousness. The author presents subliminal images as an example of decisions on sensory inputs that do not reach consciousness. The person may not have higher thinking associated with the image or describe it but they will have a certain degree of awareness that something appeared and emotional states associated to it, much like some traumas or other complex memories. Consciousness is not binary.
L363: considerably
L418: hypothetize
L414-428: There is a sort of imprecision here. Pyramidal pathways in prehensile mammals (from racoons to humans) does not 'override' completely the complex pattern generating schemes in the extrapyramidal system, but very finely adjusts them. We obviously do not think about each muscle when playing piano, but we do cue and drive each finger, until patterns have been integrated and we do not need to do it. There is no fine line between one and the other system, they are mixed, parallel. The fact that 'humans base their behavior on nonconscious processing of complex [] stimuli when subliminal' does not imply that conscience is out the window, one perpetually communicates with the other. A subliminal image can entail a fear reaction, but the person will not jump from their seat and flee the room.
L513: yet there is such a thing as code optimization and efficiency in computer programming. The complexity of an abstract task does not preclude it from the physical/mathematical rules such as that of a straight line from A to B is the fastest, and the advent of evolutionary pressure on them. I see no reason why there would not be a most efficient way to compute.
Author Response
The reviewer asks what is new in the present text compared to my previous work on consciousness. While the conclusion, that consciousness likely evolved in the early amniotes, is the same, I present additional evidence. I also suggest a model that explains how consciousness could evolve gradually as a strategy for making decisions. This angle, I believe, offers a novel insight as to what consciousness is about. Conscious decisions are just one of several strategies employed to secure survival and procreation, and apparently not the one with the larger biological success. By considering the evolutionary role of the brain, it is easier to see how advanced functions such as consciousness could evolve. It is my impression that most scientists engaged in research on consciousness doubt the model I advocate, consequently I consider it important to present supporting evidence.
As to the third comment, I certainly agree that consciousness is not a binary feature. I did discuss the point, for example, in what is now lines 121-124 (reference 6). In the revised version, the point is further elaborated by incorporating a discussion of when consciousness appears in human ontogeny.
Specific comments
L15: The point is focused on later in the text, but there is no room to expand on it in the abstract.
L71: I agree, and I do not claim it to be. In the revised version I removed one of the references to computers.
L79: Yes, I added peristaltism (or peristalsis which I believe is the more common term) as an example. As to respiration, although it can be cared for by the nonconscious, it is also under partly conscious control.
L81: The literature spans all these topics. I considered the topic too broad (and not sufficiently relevant) to expand on or cite papers.
L105: I rewrote the sentence, avoiding the term ‘designed’.
L177: There is certainly communication within a plant, as exemplified by mimosa, and there is apparently also communication between individual plants (by volatile substances released), but I agree, and stress, that I limit the term behavior to the output of nervous systems.
L255-259: I agree that it is not the complexity of feeding behavior that necessitates consciousness. I hesitate to further elaborate on the point in this part of the text, as the question is discussed in more detail later.
L277-284: I somewhat agree with the reviewer. Homo erectus was presumably reasonably successful – until the species went extinct, which, biologically speaking, is not a success. I have modified the text to stress the restricted way I use the term success.
L320-326: I have added a sentence to address the problem pointed out.
L343: As pointed out above, I certainly agree that consciousness is not either/or. (Unless one defines the term in a specific way.) The issue is relevant when discussing the transition from sleep to consciousness, the emergence of consciousness early in life, and regarding subliminal experiences. I have added a sentence to clarify the issue.
L363: The word has been corrected.
L418: The word has been corrected.
L414-428: I agree with the point made by the referee. There is a fine line between making a text comprehensible for those with limited knowledge of the topic and expanding the description in a sufficiently detailed and exact way that makes it acceptable for the specialist. That said, I have tried to improve the paragraph.
L513: Yes, but 1), I still hold that evolution not always end up with the best solution (the shortest line between A and B); and 2), conscious processing may not be even close to the optimal solution for the survival and procreation of invertebrates.
Round 2
Reviewer 2 Report
The author amended the manuscript and answered all my comments
The paragraph on ontology is welcomeI have no further comment.
Author Response
English language check has been done. Along with other revisions as to clarity and avoiding repetitive text.